# A Pilot Clinical Study to Understand the Relationship between General Movements and Ultra-Short-Term HRV of Neonates

Ziyan Wu[1], Yiming Zhong[1], Chuchu Liao[1], Xiaoyan Song[2], Qiqiong Wang[2], Wenjin Wang[1,*]

*Abstract*—Heart rate variability (HRV) reflects the regulation of the infant autonomic nervous system in neonatal care, and ultra-short-term (UST) HRV provides a faster response with higher time resolution. General movements (GMs) are indicators for the evaluation of neonatal neurological development. However, movements are often considered as a source of artifacts in HRV measurement and their physiological significance has been overlooked. This study synchronously monitors GMs and UST-HRV in a neonatal intensive care unit (NICU) to understand their physiological relationship, demonstrating the potential of UST-HRV in neonatal health monitoring. UST-HRV (a total of nine HRV parameters including RMSSD, SDNN, pNN20, LF, HF, LF/HF, SD1, SD2, SD1/SD2) is extracted from denoised electrocardiography (ECG) signals, and GMs are measured by an RGB camera with the optical flow method. Our clinical study shows that LF has a strong temporal correlation with GMs, with an average Pearson correlation coefficient of -0.623. Significant changes ($p<0.05$ for t-statistic in the linear mixed effects model) in UST-HRV are observed in SDNN, SD2, SD1/SD2, LF, HF and LF/HF before, during and after GMs. Such relationship indicates that the variation of UST-HRV is a quick and sensitive indicator for state changes such as GMs, providing an indication for clinical evaluation in dynamic events.

*Index Terms*—NICU, Health Monitoring, General Movements, Ultra-short-term HRV, ECG, Camera.

## I. Introduction

The autonomic nervous system (ANS) consists of the sympathetic nervous systems (SNS) and parasympathetic nervous systems (PNS), which counteract each other in regulating heart rate (HR), breathing rate, blood pressure, body temperature, and other functions of the body. Heart rate variability (HRV) reflects the regulation of ANS [1]. Neonatal HRV monitoring plays an important role in assessing cardiac-pulmonary regulation maturity, post-birth adaptation, pathology, and other aspects [2]. HRV measurement can be categorized into long-term

This work is supported by the National Key R&D Program of China (2022YFC2407800), National Natural Science Foundation of China (62271241, 62350068), Guangdong Basic and Applied Basic Research Foundation (2023A1515012983), Shenzhen Science and Technology Program (JSGGKQTD20221103174704003), Shenzhen Fundamental Research Program (JCYJ20220530112601003), and Shenzhen Medical Research Fund (D2402011).

[1]Department of Biomedical Engineering, Southern University of Science and Technology, China. (emails: 12212724@mail.sustech.edu.cn; 12212702@mail.sustech.edu.cn; 12331294@mail.sustech.edu.cn; wangwj3@sustech.edu.cn)

[2]Neonatal Intensive Care Unit, Nanfang Hospital of Southern Medical University, Guangzhou, China. (emails: song-xy1@163.com; 451457307@qq.com)

*Corresponding author.

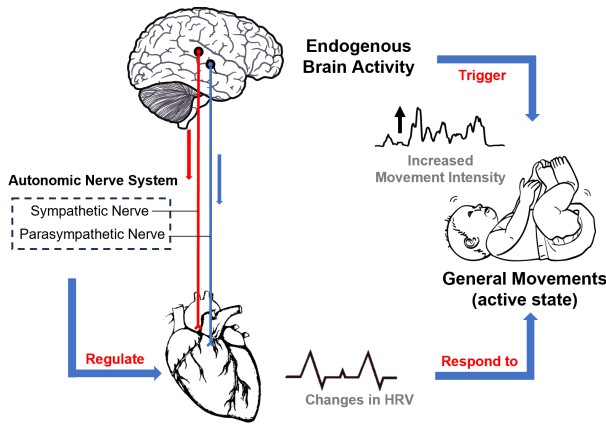

Fig. 1. The schematic diagram of the interaction between neonatal GMs and ANS regulation.

(LT), short-term (ST), and ultra-short-term (UST) monitoring. LT-HRV reflects the overall condition of the neonate, while ST-HRV reflects dynamic cardiac regulation and the heart's ability to respond to various regulatory commands [3]. However, based on current standards, a minimum measurement duration of 5 minutes is required for ST-HRV parameters in the time domain, frequency domain, and Poincare plot [4]. Due to the long measurement time, the time resolution of HRV is limited, and the clinical value is thus limited as compared to other more spontaneous physiological parameters [5]. Therefore, based on practical needs in patient monitoring, interest has grown in the measurement of UST-HRV (measurement time less than 5 minutes), as it reduces the time-latency of HRV measurement and provides quick responses with a higher time resolution. However, the relationship between UST-HRV and dynamic events such as movements is unclear since current research on UST-HRV focuses mainly on the resting state, and the UST-HRV in active states and its variations have not received sufficient attention.

Movements are important factors affecting HRV measurement [4]. In most cases, movements are considered as a source of artifacts in Electrocardiography (ECG) signal, increasing the randomness of HRV measurement [4]. Based on these artifacts, movement intensity can be quantified from ECG and Photoplethysmography (PPG) [6] [7]. However, in this view, the physiological significance of movements is overlooked. In neonatal care, spontaneous movements (SMs) refer

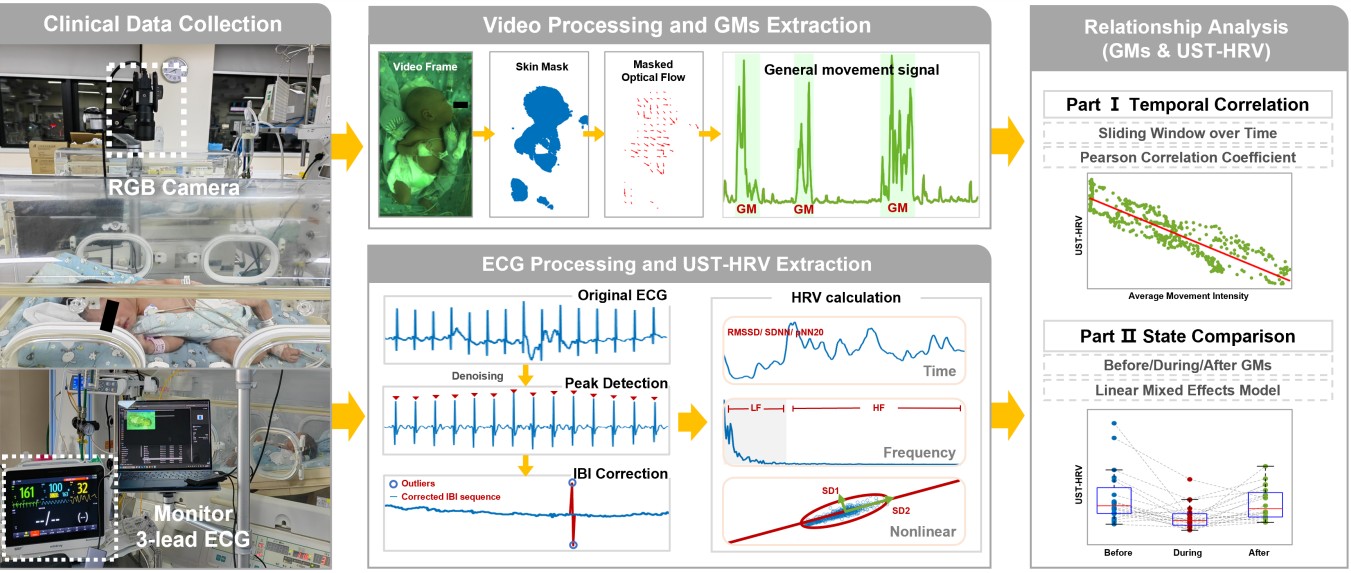

Fig. 2. The clinical setup in NICU and the workflow of camera-based GM and UST-HRV measurement.

to endogenously generated motor activities which are not related to external stimuli, and general movements (GMs) are defined as SMs involving the whole body, with a more complex pattern, and lasting from a few seconds to a minute or longer [8]. Since they are triggered by endogenous brain activity, GMs are regarded as clinical indicators related to the neurological development of newborns [8]. Additionally, GMs can influence the overall clinical state, such as neonatal respiratory variability [9]. The occurrence of GMs can be regarded as a transition from resting to active states. Such physiological fluctuation is regulated by SNS and PNS to maintain homeostasis. In terms of cardiac activity, stimulation of the SNS increases HR under high activity and stress conditions, while the PNS is responsible for decreasing HR, helping the body return to a resting state. Therefore, investigating the physiological impact of GMs on HRV can provide new perspectives for clinical decision-making. Given the timescale of GMs discussed above, UST-HRV is a suitable metric for HRV measurement.

In this study, we investigated the relationship between UST-HRV and GM by synchronously monitoring 22 infants without neurological disorders in the neonatal intensive care unit (NICU). ECG signals were recorded, denoised, and then used for UST-HRV extraction. Movements were recorded using an RGB camera and extracted by the Lucas-Kanade optical flow method [10]. The Pearson correlation coefficient (PCC) was adopted to evaluate the temporal correlation between UST-HRV and movement intensity, and the linear mixed effects model (LMM) was applied to evaluate the significance of HRV changes before, during and after GMs. The results indicate that physiologically, GMs cause the variation of UST-HRV over time. Some HRV parameters changed significantly before, during and after GMs. Our clinical study over 22 infants shows that changes in UST-HRV should not be simply considered as

random measurement errors, but rather as sensitive responses to clinical states such as GMs. Therefore, UST-HRV has the potential for real-time monitoring of neonates, providing an indication for clinical evaluation in dynamic events. The insights gained in this study also motivate further combination between UST-HRV and GMs for camera-based neonatal monitoring, considering the camera as a contactless device to record both body movements and pulse signals for HRV extraction.

## II. MATERIALS AND METHODS

### A. Clinical Setup and Data Acquisition

The study was conducted in the NICU of Nanfang Hospital of Southern Medical University, China (Approval NO. NFEC-2022-100). Under stable illumination, the raw videos were recorded with an RGB camera (IDS-Ul3860C, Germany) with a resolution of $484 \times 274$ pixels, sampled at 60 FPS. The camera was mounted directly above the infant, filming the entire body.

The three-lead ECG was synchronously recorded using the Benevision N17 patient monitor (Mindray, China), sampled at 500 Hz. The PPG was not considered as it is severely affected by motion artifacts.

Infants in incubators were recorded. After manually removing samples interfered with clinical operations such as feeding or changing diapers, a total of 22 infants were recorded, with an average duration of 10 minutes for each. Table I shows the information of infants recorded, detailing gender, postmenstrual age (PMA) and diagnosis. Although all infants were under medical observation, none of them had neural disorders.

### B. UST-HRV extraction

In ECG processing, motion artifacts primarily consist of short-term baseline wanders and electromyographic (EMG)

TABLE I
THE INFORMATION OF INFANTS RECORDED

| ID | Gender | PMA | Diagnosis | ID | Gender | PMA | Diagnosis |
|---|---|---|---|---|---|---|---|
| 1 | Female | 31 | Low Birth Weight | 12 | Female | 28 | Preterm |
| 2 | Male | 34 | Neonatal Jaundice | 13 | Male | 31 | Low Birth Weight |
| 3 | Female | 38 | Acidosis in Blood Gas Analysis | 14 | Female | 31 | Low Birth Weight |
| 4 | Female | 34 | Low Birth Weight | 15 | Female | 40 | Meconium-Stained Amniotic Fluid, Grade 3 |
| 5 | Male | 35 | Meconium-Stained Amniotic Fluid, Grade 3 | 16 | Male | 36 | Intrapartum Fetal Distress |
| 6 | Female | 32 | Preterm | 17 | Female | 33 | preterm |
| 7 | Male | 32 | Preterm | 18 | Male | 32 | Preterm |
| 8 | Male | 34 | Preterm | 19 | Male | 34 | Low Birth Weight |
| 9 | Female | 30 | Low Birth Weight | 20 | Female | 32 | Preterm |
| 10 | Male | 34 | Preterm | 21 | Female | 33 | Low Birth Weight |
| 11 | Male | 28 | Neonatal Respiratory Distress Syndrome | 22 | Male | 32 | Preterm |

activities, with an average amplitude typically being 10% of the peak-to-peak ECG amplitude [11]. Bandpass filtering (normally 0.05-40 Hz [11]) can retain the ECG signal while filtering out some EMG components, and baseline wanders can be effectively eliminated using wavelet transform [12]. The ECG acquired from the Mindray monitor was decomposed into 11 levels using discrete Meyer wavelets, and levels 1 to 7 were retained [13], followed by a bandpass filter with a cutoff frequency of 0.05-35 Hz to further filter out EMG components. Bidirectional filtering was implemented using the MATLAB function filtfilt(·) to avoid any phase shifts.

The MATLAB function findpeaks(·) was applied to detect R-peaks in the ECG signal. Based on the prior knowledge of the average signal amplitude of our data and the normal HR range (120-200 bpm) in newborns, two threshold parameters MinPeakHeight and MinPeakDistance were set to filter out invalid peaks.

The inter-beat intervals (IBI, intervals between two adjacent R-peaks) were corrected based on the fractional difference between current IBI and the previous two IBIs. If the difference exceeds a threshold of 25%, the current IBI will be considered as a candidate of outliers. If the IBI falls out of the range of mean ± 3 times the standard deviation of all IBIs, it will be replaced by the median value of the nearest 7 IBIs.

A total of 9 HRV parameters were extracted from time domain (RMSSD, SDNN, pNN20), frequency domain (LF, HF, LF/HF), and Poincare plot (SD1, SD2, SD1/SD2). The parameter pNN50 was replaced by pNN20 for neonatal HRV analysis due to the high HR of neonates [14]. These 9 HRV parameters are regular metrics used frequently in clinics (e.g. Mindray BeneVision patient monitors) with sufficient medical studies, covering common analytical methods. Although a more comprehensive set of HRV parameters could theoretically be incorporated, our primary objective is to demonstrate dynamic changes in UST-HRV over dynamic events. We suggest that the most regular and widely adopted indicators suffice for our research purposes.

For the frequency domain parameters, the IBI sequence was firstly resampled to 8 Hz and then the Welch's periodogram method was applied to estimate the power spectral density (PSD). Finally, the LF and HF are respectively defined as absolute power in 0.04–0.2 Hz and 0.2–1.4 Hz, which are recommended for neonatal populations rather than the standard adult ranges [4].

### C. GMs Extraction

Movements were quantified from the video using the Lucas-Kanade optical flow method [10]. Firstly, the optical flows of the entire video frame were extracted. Then, in order to eliminate the interference of the background region, the OTSU method was applied to segment the skin area [15]. Finally, the total amplitude of all the optical flows in the skin area was used as the general body movement signal.

Based on the manual video examination and the amplitude of optical flows, GM events were labeled under the guidance of medical experts. Given that extremely short movements are more likely to be isolated movements or reflexive behaviors, only movement segments longer than 20 seconds were considered as GMs [16].

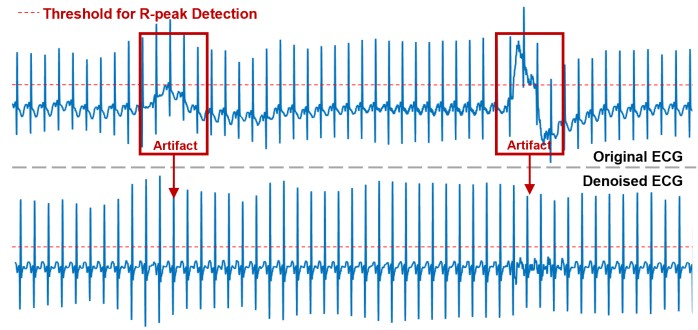

Fig. 3. Example of ECG signals obtained GMs in our dataset. The motion artifacts in the original ECG signal are eliminated in the de-noised ECG signal.

### D. Analysis of relationships between UST-HRV and GMs

Fig. 3 exemplifies ECG signals obtained GMs in our dataset. We have very carefully checked the moments where GMs occur. The denoised ECG signal is not distorted by motion, which ensures that the R-peak detection works properly to produce results that are physiologically induced by GMs rather than artifacts.

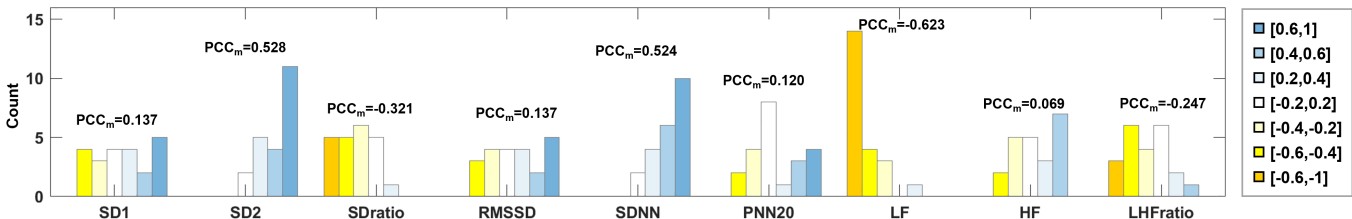

Fig. 4. The PCCs of different HRV parameters with GMs. $PCC_m$ denotes the mean of PCCs of the HRV parameter.

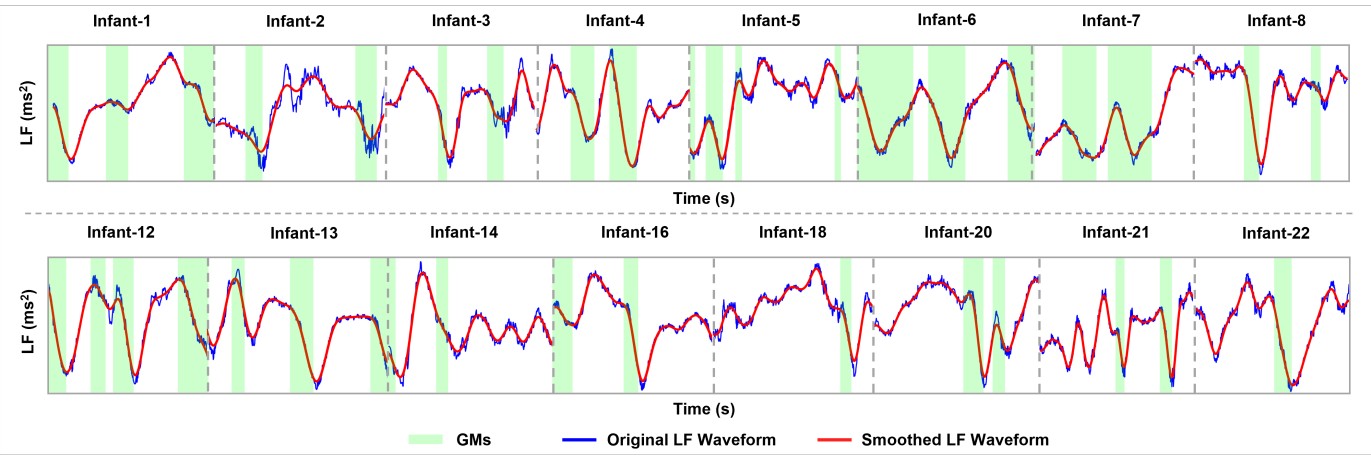

Fig. 5. Occurrence of GMs in the synchronous monitoring of LF. The LF trace is smoothed using a 15-second sliding average window. Generally when GMs occur, LF decreases. Each GM causes a valley in the corresponding LF waveform.

*1) Temporal Correlation:* For each infant, a sliding window of 1 minute was adopted to continuously extract UST-HRV and the average movement intensity. The Pearson correlation coefficient (PCC) was calculated to evaluate their temporal correlation. The absolute value of PCC was defined as follows: 0-0.2 as no correlation, 0.2-0.4 as weak correlation, 0.4-0.6 as moderate correlation, and above 0.6 as strong correlation. The number of samples in each interval for each HRV parameter was counted to reflect the overall correlation. Based on the PCC results, the parameter LF was chosen as a reference to visually demonstrate how GMs align with real-time HRV waveforms.

*2) State Comparison:* The UST-HRV parameters before, during and after GMs were compared using a linear mixed-effects model (LMM). LMM is a flexible model that combines fixed effects (GMs) and random effects (inter-subject differences) to evaluate non-independent data and has been used in medical research [17]. The built-in MATLAB function fitlme(·) was applied to establish the LMM and provide the t-statistic to measure the differences in UST-HRV under different states. For the modeling of random effects, we only adopted random intercepts to capture baseline variations across infants. While random slopes captures variations of predictor effects, they increase model complexity, thus not adopted due to limited data and overfitting concerns.

To ensure that infants were in the resting state before GMs, the selected GMs in the comparison met the following criterion: there were no other GMs or excessive isolated movements occurring two minutes before the onset of the selected GMs. For each GM event, we analyzed:

- 1-minute preceding movement onset (Before).
- 1-minute aligned with the center of GM (During).
- 1-minute following movement cessation (After).

## III. RESULTS AND DISCUSSION

### A. Temporal Correlation between UST-HRV and GMs

Fig. 4 shows statistical results of PCC between different HRV parameters and GMs, which are characterized by high movement intensity.

SDNN (mean PCC = 0.524) and SD2 (mean PCC = 0.528) have a relatively strong positive correlation with GMs. Short-term SDNN mainly reflects the short-term fluctuations in HR, with both SNS and PNS contributing to it [18]. Although the SNS activity leads to a reduction in HRV [18], SDNN can increase or decrease as a result of decrease in HRV [19]. In our study, the brief duration of GMs, along with the rapid transition between resting and active states, causes a rapid change in HR, which in turn leads to an increase in SDNN during the presence of GMs. SD2 is equivalent to SDNN both in statistical meanings and medical significance [18] and therefore has a similar result.

LF (mean PCC = -0.623) has a strong, negative correlation with GMs. LF is not specifically associated with any particular

## TABLE II
### STATISTICAL RESULTS OF LMM COMPARISON

| HRV | States | p-value | 95% CI | HRV | States | p-value | 95% CI | HRV | States | p-value | 95% CI |
|---|---|---|---|---|---|---|---|---|---|---|---|
| SD1 | B-A | 0.060 | [-0.090225, 4.1068] | SD2 | B-A | 0.109 | [-1.6364, 3.1498] | SD1/SD2 | B-A | 0.197 | [-13.124, 2.7653] |
| SD1 | B-D | 0.320 | [-1.0472, 3.1498] | SD2 | B-D | <0.001 | [14.818, 32.229] | SD1/SD2 | B-D | <0.001 | [-24.476, -8.5872] |
| SD1 | D-A | 0.365 | [-3.0554, 1.1415] | SD2 | D-A | <0.001 | [7.7491, 25.163] | SD1/SD2 | D-A | 0.006 | [-19.297, -3.4078] |
| RMSSD | B-A | 0.060 | [-0.12575, 5.7805] | SDNN | B-A | 0.092 | [-0.87347, 11.162] | pNN20 | B-A | 0.255 | [-2.2911, 8.4791] |
| RMSSD | B-D | 0.319 | [-1.4695, 4.4367] | SDNN | B-D | <0.001 | [10.167, 22.202] | pNN20 | B-D | 0.591 | [-6.8388, 3.9313] |
| RMSSD | D-A | 0.366 | [-4.2969, 1.6094] | SDNN | D-A | <0.001 | [5.0229, 17.058] | pNN20 | D-A | 0.096 | [-9.9329, 0.83719] |
| LF | B-A | 0.162 | [-22810, 4319.4] | HF | B-A | 0.046 | [-3.5126, 82.023] | LF/HF | B-A | 0.055 | [-35596, 509.08] |
| LF | B-D | 0.041 | [-28041, -911.44] | HF | B-D | 0.347 | [-46.874, 111.66] | LF/HF | B-D | 0.006 | [-49690, -9727] |
| LF | D-A | <0.001 | [-37286, -10157] | HF | D-A | 0.281 | [-67.627, 20.907] | LF/HF | D-A | 0.370 | [-36075, 7887.5] |

Column "States": B-Before GM, A-After GM, D-During GM.

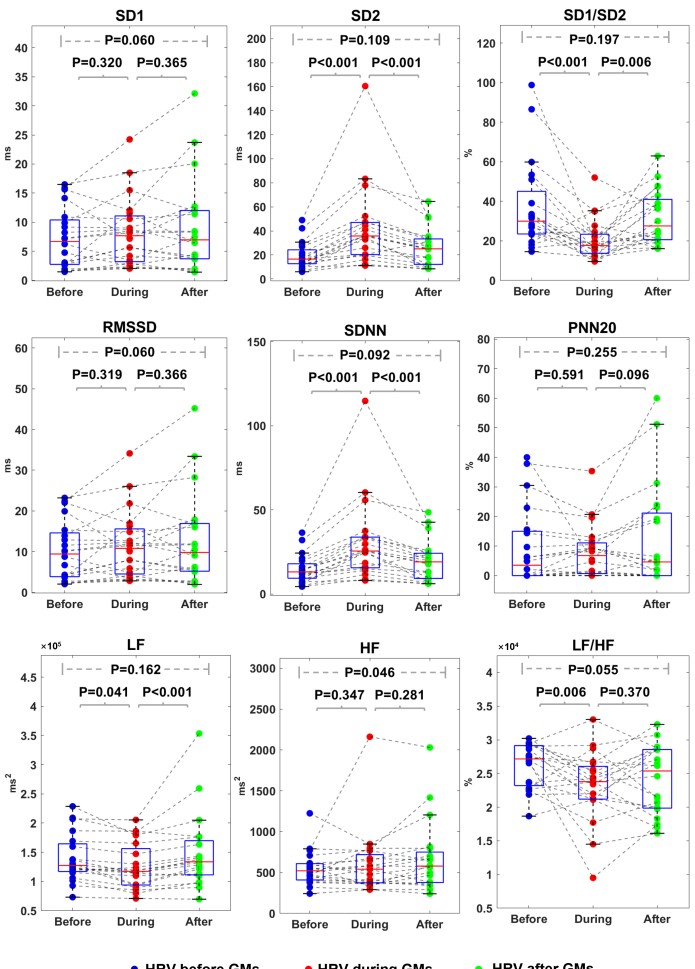

Fig. 6. Overview of ANS regulation in GMs along with UST-HRV changes.

neurological activity, but rather reflects the overall influence of SNS, PNS, and baroreceptor activities [3]. It reflects the complex and overall regulation of the ANS [3]. Fig. 5 shows the occurrence of GMs in the synchronous monitoring of LF. When GMs occur, LF decreases, leading to valleys of the waveform. Specifically, all samples with a low correlation in LF have a common characteristic: they have no GMs that last for at least 20 seconds, which indicates a strong relationship between LF and GMs. This may suggest that movements cause a significant change in the ANS regulation only when they last long enough.

### B. State Comparison Before, During and After GMs

Fig. 6 summarizes changes in UST-HRV before, during and after GMs, with Fig. 7 illustrating data distribution and Table II showing statistical results. Before the occurrence of GMs, the infants are in a resting state, which serves as the reference state for comparison. Compared to the resting state before GMs, the focus is on how GMs induce differences in UST-HRV during and immediately after their occurrence.

During GMs, SDNN, SD2 and LF are significantly different (p<0.05 for t-statistic in LMM) from those before and after GMs. SDNN and SD2 increase while LF decreases. The result is consistent with the correlation between GMs and UST-HRV discussed in the previous sections, which again indicates that the overall variability of HR and the ANS regulation changes during GMs, and changes in these three parameters are highly related to GMs. In addition, SD1/SD2 has the same significant variation pattern as LF. SD1/SD2 describes the sympathovagal balance, a measure of the relative contributions of SNS to PNS activity [20] and has been shown to additionally capture the complexity in HR patterns [18]. Its variation indicates that

Fig. 7. Differences of HRV parameters before, during and after GMs. The boxplot illustrates the statistical distribution of HRV parameters under specific GM states. Dashed lines connect different GM states of a same infant. The p-value measures the significance of differences among different states.

specific HR patterns are generated due to GMs. Finally, LF/HF during GMs is significantly different from that before GMs. It also measures the sympathovagal balance [18], capturing changes in ANS regulation induced by GMs.

After GMs, HF differs significantly from that before GMs.

Unlike parameters that describe the overall ANS regulation, HF primarily reflects the activity of the PNS [18]. After GMs, to reduce HR and regulate the body back to a resting state, PNS dominates in the ANS and thus its activity increases. This regulation pattern is captured by the increased HF.

Considering the correlation between GMs and UST-HRV discussed previously, SDNN, SD2, and LF exhibit a higher degree of sensitivity to such state alternations, making them more promising for ultra-short-term analysis. While other HRV parameters, though still valuable, require further investigation to fully understand their role and potential.

## IV. Conclusion

In this study, the physiological relationship between real-time UST-HRV and neonatal GMs is investigated. LF has a strong temporal correlation (mean PCC = -0.623) with GMs, while SDNN and SD2 have a relatively strong correlation (mean PCC = 0.524, 0.528, respectively). Significant changes ($p<0.05$) in SDNN, SD2, LF, HF, LF/HF and SD1/SD2 are able to characterize the changes in the overall ANS regulation before, during and after GMs. The variation in UST-HRV is a sensitive response to clinical states such as GMs and valuable for real-time infant monitoring in NICU, which may provide new insights for patient monitoring and care.

Our future work aims to develop a camera-based monitoring system that integrates UST-HRV with GMs to provide a full-fledged solution for infant health tracking. In addition, to explore further clinical applicability, a broader range of clinical conditions with physiological value could be considered, such as measuring UST-HRV and GMs during infant sleep, crying, and apnoea.

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
