# OpenReview forum: "A Pilot Clinical Study to Understand the Relationship between General Movements and Ultra-Short-Term HRV of Neonates"
_IEEE.org/EMBS/BHI/2025/Conference — BHI 2025_

### Official Review · Reviewer_6hFn · 2025-07-13
**This pilot study explores the relationship between general movements (GMs) and Ultra-Short-Term heart rate variability (UST-HRV) in neonates using synchronized ECG and camera-based motion tracking.  The clinical motivation is valuable, however, the scientific contribution is limited due to small sample size, lack of predictive modeling, and primarily descriptive analysis.**

**Confidence:** 4
**Clarity Of Writing:** good
**Clinical Significance:** fair
**Methodological Novelty:** fair
**Overall Rating:** 2

**Experiments And Results:**

fair

**Questions For The Authors:**

1. You applied UST-HRV feature extraction using a sliding window of 1 minute to assess temporal correlation between HRV and GMs. However, during state comparisons, it’s unclear over what duration HRV features were calculated for the “before”, “during”, and “after” GM segments. Could you clarify the length and overlap (if any) of those segments? Since frequency-domain HRV parameters (e.g., LF, HF) typically require ≥60s duration for reliable PSD estimates, how do you justify using them in shorter segments, especially during brief GMs?
Impact: The reliability of spectral HRV metrics in ultra-short (<1 min) segments is debated; clarification would affect the credibility of physiologic interpretations.

2. The paper defines GM segments as body movements lasting over 20 seconds, based on amplitude of optical flow and manual inspection. Could you elaborate on how this 20s threshold was chosen? Were alternative thresholds (e.g., 10s or 30s) tested for robustness? Additionally, was there any validation against gold-standard annotations (e.g., trained neurophysiologist labelings) to confirm GM identification?
Impact: The movement detection directly influences both HRV alignment and group comparison validity.

3. You excluded GM segments that had other GMs or “excessive isolated movements” two minutes prior. Could you clarify how these isolated movements were quantified and why they were not formally analyzed for their own effect on HRV? Would comparing HRV during isolated vs. GM movements help understand motion-induced HRV modulation better?
Impact: It may reveal whether GM-induced HRV changes are unique, or part of a broader pattern.

4. (1) Why did you choose to compute the specific set of HRV metrics (e.g., LF, SDNN, SD2, SD1/SD2)? Have you explored additional features such as entropy-based metrics, DFA (detrended fluctuation analysis), or Poincaré geometric descriptors that are less sensitive to non-stationarity and more suitable for ultra-short HRV?
Impact: If alternative features perform better under ultra-short timeframes, they may provide more stable and interpretable signals, particularly in high-noise NICU settings.
(2) You extracted nine HRV metrics including LF, HF, SDNN, RMSSD, etc., and redefined pNN50 as pNN20 for neonates. Could you comment on whether all selected metrics are validated in the neonatal population, particularly LF/HF and SD1/SD2, whose physiological meaning may differ from adults?
Impact: The interpretability of sympathovagal balance metrics in neonates may not be straightforward without developmental normalization.

5. You applied a linear mixed-effects model (LMM) to evaluate UST-HRV differences across states. Could you clarify how the random effects were structured (e.g., intercept-only per infant)? Was the potential within-subject autocorrelation of HRV features over time modeled explicitly? Given that HRV is known to have temporal dependency, this could influence the t-statistics used to support significance.
Impact: Model mis-specification may inflate significance, weakening the strength of reported findings.

6. The Pearson correlation between HRV and motion intensity was computed using a sliding window. Was any correction performed for autocorrelation or potential shared trends (e.g., both HRV and movement rising over time)? Otherwise, Pearson correlation may spuriously reflect co-trending rather than causal association.
Impact: Correlation may overstate physiological linkage if confounding temporal structures are not accounted for.

7. The study includes 22 neonates, but there is no mention of confidence intervals or effect sizes when reporting significant changes in HRV. Could you provide CI estimates for HRV parameter changes before/during/after GMs, to help assess variability and generalizability?
Impact: P-values alone are insufficient for assessing the robustness and clinical meaningfulness of observed effects.

8. Any HRV → GM Prediction?
The paper avoids using HRV to predict GM occurrence. Given that HRV computation is real-time and GMs are downstream events, why didn’t you consider constructing a classifier (even logistic regression or simple thresholding) to predict GM presence or onset based on HRV features?
Impact: Even a simple classification analysis would elevate this work from observational to clinically actionable, helping establish feasibility of real-time neuromotor monitoring.

**Strengths:**

1. Innovative use of UST-HRV in dynamic states: Most HRV research focuses on resting state, and short-term HRV (≥5 minutes). This paper addresses UST-HRV changes before, during, and after GMs, aligning with the short timescale of GMs and making a case for high-resolution physiological monitoring in neonatal care (page 1).
2. Rigorous signal processing pipeline: The ECG signals were denoised using discrete Meyer wavelet decomposition (levels 1–7 retained), followed by 0.05–35 Hz bandpass filtering and zero-phase filtfilt() filtering, minimizing EMG artifacts. The processed signal allowed reliable R-peak detection with physiological constraints on HR (120–200 bpm) and IBI correction thresholds (25% deviation, 3σ rule). These careful preprocessing steps ensure robustness in noisy NICU environments.
3. Well-constructed GM quantification using optical flow and OTSU skin segmentation: GM activity was extracted using Lucas-Kanade optical flow, and only movements longer than 20 seconds were labeled as GMs under clinical supervision. This avoids confounding with short reflexes or isolated twitching, ensuring data integrity.
4. Statistical analysis is appropriate: Temporal correlation via Pearson correlation coefficient (PCC) revealed LF had a strong negative correlation with GMs (mean PCC = –0.623), while SDNN and SD2 showed moderate positive correlation (≈0.52) (Fig. 4–5). A linear mixed effects model (LMM) was applied to analyze HRV changes before/during/after GMs, showing significant (p < 0.05) state-dependent changes in SDNN, SD2, LF, LF/HF, and SD1/SD2 (Fig. 6–7), adding statistical credibility.

**Summary Of The Paper:**

The paper investigates the relationship between GMs and UST-HRV in a neonatal intensive care unit (NICU) using data from 22 patients.
Methods: It synchronizes ECG recordings and RGB video to extract HRV parameters and quantify movement via optical flow.
Temporal correlations and a linear mixed-effects model are used to assess HRV parameter changes before, during, and after GMs.
Results: The paper reports significant differences in LF, SDNN, SD2, and other HRV metrics across states, suggesting that UST-HRV can serve as a sensitive physiological marker.

**Weaknesses:**

1. Lack of generalizability and statistical power: The study only includes 22 neonates from a single NICU, all without neurological disorders. This makes the result less generalizable, particularly for infants with abnormal development—the group for whom GM-related HRV would be most clinically valuable.

2. No hypothesis testing beyond statistical correlation:
The paper avoids forming any mechanistic or causal hypotheses. Although LF shows a significant drop during GMs (only a correlation), the authors do not explore why LF (a mix of SNS/PNS/baroreflex activity) drops specifically, or how GMs neurologically modulate ANS from biological perspective and mechanistic insight. This weakens scientific interpretability, especially for this kind of clinical translation research.

3. No predictive or classification modeling to link HRV with outcomes or diagnosis:
The study stops at descriptive statistics. There is no attempt to classify GM occurrence based on HRV (or vice versa), or to propose how this system could aid clinical alerts. No machine learning or thresholding pipeline is proposed for real-world use.

4. No multi-center data or external replication: All data were collected under a single protocol at one hospital. No discussion is given for how well this approach generalizes to different equipment, lighting, infant states, or motion capture systems.

5.Clinical relevance is vague: Although the authors emphasize HRV as a “sensitive marker,” no evidence is presented that these signals help predict adverse events, developmental delays, or medical interventions.

---

### Official Review · Reviewer_aq1r · 2025-07-16
**A Pilot Clinical Study to Understand the Relationship between General Movements and Ultra-Short-Term HRV of Neonates**

**Confidence:** 5
**Clarity Of Writing:** great
**Clinical Significance:** good
**Methodological Novelty:** good
**Overall Rating:** 7

**Experiments And Results:**

good

**Questions For The Authors:**

No specific question to authors

**Strengths:**

- The methodology is clear and correctly described

**Summary Of The Paper:**

In this paper, authors propose to observe the correlation between general movements and HRV features in neonates.
They show a link between GMs and HRV features and the methodology is pertinent and correctly described.

**Weaknesses:**

- last paragraph of the introduction contains all the content as an abstract. Only the "broad" plan should be detailed
- The way of producing the Figure 6 could be better explain. Is it observed for all babies ? Is it a global trends?
- Dataset is small and may be unbalanced. It would have been interesting to known the repartition of GMs events in the different recording
- Is it possible that findings are related to PMA ? 22 newborns going from 26 to 40 weeks (quite wide PMA range) but may be not enough to cover it.
- Figure may not be cited in the text.

---

### Official Review · Reviewer_ia6j · 2025-07-17
**Well-designed pilot study shows promising link between neonatal movement and ultra-short-term HRV**

**Confidence:** 4
**Clarity Of Writing:** good
**Clinical Significance:** good
**Methodological Novelty:** great
**Overall Rating:** 7
**Final Rating:** 8

**Experiments And Results:**

great

**Questions For The Authors:**

1. Would the HRV parameters behave similarly in neonates with abnormal neurodevelopment or pathological conditions?
2. How might the proposed method scale to a fully camera-based system without ECG input?

**Strengths:**

1. Uses synchronized ECG and video monitoring to capture physiological and behavioral data in real clinical settings.
2. Clearly defines UST-HRV metrics.
3. Applies rigorous statistical analysis (LMM) to handle repeated measurements and inter-subject variability.

**Summary Of The Paper:**

The paper investigates how general movements (GMs) relate to ultra short term heart rate variability (UST HRV) in neonates in a neonatal intensive care unit. It synchronously records three-lead ECG signals and RGB video for 22 infants with no neurological disorders, extracts nine UST HRV parameters from denoised ECG, and quantifies GMs via optical flow and manual labeling. Temporal correlation is assessed with sliding-window Pearson coefficients, and state changes before, during, and after GMs are evaluated using a linear mixed effects model. The study finds that low-frequency power shows a strong negative correlation, while SDNN and SD2 show moderate positive correlations, and that multiple UST-HRV metrics change significantly across GM states.

**Weaknesses:**

1. Limited sample size.
2. Only general movements lasting more than 20 seconds were analyzed. Shorter movements, or micro-movements, potentially clinically relevant, were excluded.

---

### Official Review · Reviewer_Qf32 · 2025-07-18
**Solid association analysis between UST HRV and GM with classical statistical methods**

**Confidence:** 3
**Clarity Of Writing:** excellent
**Clinical Significance:** great
**Methodological Novelty:** good
**Overall Rating:** 7

**Experiments And Results:**

excellent

**Questions For The Authors:**

1. Please provide a demographics table for the 22 infants.
2. In methods, elaborate more on the rationale for choosing those 8 HRV features.
3. Implement an automatic strategy for GM event identification and re-perform the analysis shown in Figure 7. This would reduce potential bias introduced by manual clinician labeling and strengthen the objectivity and reproducibility of the conclusions.

**Strengths:**

1. The paper addresses a clinically relevant topic: how spontaneous movements may associate with HRV.

2. The combination of camera-based movement quantification and ECG-based HRV analysis are both innovative and supports non-contact monitoring.

3. Methodological Rigor: The use of wavelet denoising, solid correlation analysis is well justified and appropriate for the pilot scale.

**Summary Of The Paper:**

This is a pilot study trying to understand the association between ultra-short-term heart rate variability (HRV) and general movement (GM). They recruited 22 infants. Then, they obtained the ECG data from the children and perform sequence analysis to obtain the RR interval to calculate 9 different measures of HRV. The GM extraction is obtained from the camera with OTSU image processing method. They demonstrate a clear assocation between HRV signals and the GMs.

**Weaknesses:**

1. It would be beneficial for authors to explain why you choose 9 HRV features. What does each of them represent different aspect of HRV features.
2. The definition of GM events is dependent on the medical experts' decision. However, in Fig 2, it seems to be pretty clear that maybe based on some automatic time series processing strategy. For example, thresholding, or sum of movement signal over time. Creating an automatic strategy to identify the GM events can help to eliminate the subjectivity in defining the GM events from clincian's, and thus making the state comparison analysis more faithful.